# Intracranial Inflammatory Myofibroblastic Tumor: A Literature Review and a Rare Case Misdiagnosed as Acoustic Neuroma

**DOI:** 10.3390/diagnostics13172725

**Published:** 2023-08-22

**Authors:** Le Zhou, Wanqian Pan, Renjun Huang, Ziwei Lu, Zhiqun You, Yonggang Li

**Affiliations:** 1Department of Radiology, The First Affiliated Hospital of Soochow University, Suzhou 215000, China; zhoule1113@163.com (L.Z.); 729035422@163.com (R.H.); eggless@126.com (Z.L.); 2Department of Cardiology, The First Affiliated Hospital of Soochow University, 188 Shizi Street, Suzhou 215006, China; 1730805144@stu.suda.edu.cn; 3Department of Pathology, The First Affiliated Hospital of Suzhou University, Suzhou 215000, China; 460248728@163.com; 4Institute of Medical Imaging, Soochow University, Suzhou 215000, China; 5National Clinical Research Center for Hematologic Diseases, The First Affiliated Hospital of Soochow University, Suzhou 215000, China

**Keywords:** inflammatory myofibroblastic tumor, MRI, acoustic neuroma, surgical resection, headache

## Abstract

Inflammatory myofibroblastic tumor (IMT) stands as a rare neoplasm, initially documented by Bahadori and Liebow in 1973; however, its biological behavior and underlying pathogenesis continue to elude comprehensive understanding. Throughout the years, this tumor has been designated by various alternative names, including pseudosarcomatoid myofibroblastoma, fibromyxoid transformation, and plasma cell granuloma among others. In 2002, the World Health Organization (WHO) officially classified it as a soft tissue tumor and designated it as IMT. While IMT primarily manifests in the lungs, the common clinical symptoms encompass anemia, low-grade fever, limb weakness, and chest pain. The mesentery, omentum, and retroperitoneum are subsequent sites of occurrence with intracranial involvement being exceedingly rare. Due to the absence of specific clinical symptoms and characteristic radiographic features, diagnosing intracranial inflammatory myofibroblastic tumor (IIMT) remains challenging. Successful instances of pharmacological treatment for IIMT indicate that surgery may not be the sole therapeutic recourse, thus underscoring the imperative of an accurate diagnosis and apt treatment selection to improve patient outcomes.

## 1. Introduction

IMT is a rare junctional/low-grade tumor with the potential for recurrence and progression [1]; it is mainly composed of myofibroblast spindle cells with inflammatory cell infiltration, including plasma cells, lymphocytes, and eosinophils [2]. In 2002, the WHO officially defined it as a soft tissue tumor and named it IMT [3]. The occurrence of IMT may be related to numerous factors, such as Epstein–Barr virus infection [4], surgery, trauma, etc. In addition, rearrangement and overexpression of the anaplastic lymphoma kinase (*ALK*) gene on chromosome 2p23 are also associated with IMT [5]. In 2000, Lawrence et al. found that the TPM3–ALK and TPM4–ALK fusion genes were related to IMT [6]. Since then, numerous ALK fusion genes have been reported, such as ATIC–ALK, CLTC–ALK, CARS–ALK, RANBP2–ALK, RRBP1–ALK, etc. [5,7,8]. In IMT, approximately 50% of *ALK* genes are rearranged and overexpressed; therefore, ALK positivity is helpful to diagnose IMT, but its absence does not exclude the diagnosis of IMT, particularly in adults [9,10]. The characteristic histological patterns include the fasciitis-like, compact spindle cell and hypocellular fibrous patterns [7,9]. While the most common site of occurrence is the lungs, extrapulmonary IMT tends to exhibit a more aggressive behavior and a higher susceptibility to recurrence [11]. The first reported case of IIMT was presented by SG West et al. in 1980 [12]. IIMT is extremely rare, and its clinical presentation often lacks a correlation with its histological type. Patients typically present with non-specific symptoms, such as headaches and seizures among others [13]. The preoperative diagnosis of IIMT poses challenges. Previous radiological studies have primarily depicted IIMT as a solid mass; however, Wang [14] and Park [15], among others, have also identified instances of cystic IIMT. This substantiates the notion that there is no distinct morphological component characteristic of IIMT evident in radiographic imaging.

The current body of literature concerning IIMT primarily comprises case reports, thus lacking a comprehensive overview. This article presents the first instance of hemorrhagic IIMT treated at the First Affiliated Hospital of Soochow University, which was initially misdiagnosed as an acoustic neuroma. Additionally, we conducted a comprehensive review of the relevant literature on IIMT from 1980 to 2022 using the PubMed database, aiming to enhance our understanding of this rare neoplasm.

## 2. Case Presentation

On 27 July 2022, a 55-year-old male presented at the Second People’s Hospital in Changshu, Jiangsu Province, China with unexplained symptoms of dizziness, unsteady walking, choking on water, and hoarseness. The attending doctor initially diagnosed an acoustic neuroma in the right cerebellopontine angle. Subsequently, the patient sought further treatment at our hospital. The patient had a history of hypertension for over 20 years but had stopped taking medication on his own for three years, perceiving stable blood pressure. Additionally, he had a history of radiotherapy for nasopharyngeal cancer 20 years ago and had been regularly followed up after discharge. Routine laboratory tests showed increased neutrophils, leukocytes, and C-reactive protein with other values within normal limits. The chest computed tomography (CT) examination revealed no abnormalities.

Cranial magnetic resonance imaging (MRI) demonstrated a circular mixed T1-weighted imaging (T1WI) and T2-weighted imaging (T2WI) signal shadow in the right cerebellopontine angle with a lesion size of approximately 3.9 × 3.2 × 3.0 cm, mixed diffusion-weighted imaging (DWI) and apparent diffusion coefficient (ADC) signals, and heterogeneous enhancement. The compression of the right cerebellar hemisphere and the four ventricles resulted in their displacement towards the left (Figure 1). The neuroradiologist at our hospital also diagnosed it as an acoustic neuroma. The patient underwent tumor resection, and histopathological analysis revealed a lesion comprised of fibrous collagen tissue with hemorrhagic infarcts and lamellar capillary hyperplasia. Within the lesion, there were perivascular spindle cells and epithelioid proliferation, exhibiting cellular heterogeneity and easily visible nuclear pleomorphism. Immunohistochemical staining demonstrated negativity for glial fibrillary acidic protein (GFAP), S100 protein, Oligodendrocyte transcription factor 2 (Olig-2), CD56, SOX10, and epithelial membrane antigen (EMA). Additionally, the lesion was negative for estrogen receptor (ER), progesterone receptor (PR), somatostatin receptor 2 (SSTR2), cytokeratin (CK), desmin, and CD68. The lesion displayed positivity for vimentin and exhibited scattered weak positivity for smooth muscle actin (SMA) in the vascular regions (Figure 2). Furthermore, immunostaining revealed CD34 and CD31 positivity in the vascular regions as well as erythroblast transformation specific related gene (ERG) and Ki-67 (approximately 40% in the hotspot areas) positivity (Figure 2). Based on these findings and the WHO classification guidelines (Table 1, the postoperative pathology confirmed an intracranial inflammatory myofibroblastic tumor.

Approximately one year after hospital discharge, we conducted a follow-up via phone call, during which the patient reported no notable discomfort or concerns.

## 3. Literature Review

This misdiagnosed case prompted us to conduct an in-depth investigation into IIMT. Due to the historical use of various names for this tumor, we conducted a comprehensive search of the PubMed database using the following keywords: “inflammatory myofibroblastic tumor,” “intracranial plasma cell granuloma,” “inflammatory pseudotumor,” “cellular inflammatory pseudotumor,” “fibrous yellow tumor,” “yellow tumor pseudotumor,” “pseudosarcomatous myofibroblast proliferation,” and “inflammatory myofibrohistiocytic proliferation”. We selected articles encompassing case reports, reviews, and other relevant studies for analysis. The literature was carefully reviewed to extract the pertinent information, such as the tumor’s location of occurrence, clinical manifestations, treatment modalities, and more. The results are shown in Table 2.

After excluding literature with unavailable content, a total of 55 cases were included in the study until October 2022. The age range of patients varied from 6 to 82 years with 23 females (41.8%) and 32 males (58.2%). While IMT primarily occurs in the lung and respiratory tract, it has also been reported in various other sites, such as the breast, esophagus, intestine, kidney, liver, lymph nodes, retroperitoneum, oral cavity, skin, stomach, thyroid, etc. However, the occurrence of IMT within the intracranial region remains exceptionally rare. From Table 2, we observed that IIMT can arise in almost all intracranial locations with the frontal (23.6%) and temporal lobes (21.8%) being the main sites. The clinical symptoms commonly included headache (56.4%) and seizures (18.2%) with headache being the predominant complaint. A few cases presented with metastasis (three cases) and recurrence (ten cases) with the earliest recurrence noted six months after treatment, and the latest recurrence occurring in the 11th year. Regarding the treatment of IIMT, various strategies were employed, including surgery, glucocorticoids, radiotherapy, chemotherapy, immunosuppressants, 6-mercaptopurine, methotrexate, non-steroidal anti-inflammatory drugs (NSAIDs), thalidomide, and amphotericin. From Table 2, we observed that most cases exhibited iso or low signal intensity on T1WI and T2WI with the DWI signal usually being low. Moreover, most of the lesions showed significant enhancement (although some were not homogeneous). Table 2 also lists the initial diagnoses made by the clinicians and neuroradiologists before the pathological results were obtained, based only on the radiological features and clinical symptoms. Interestingly, meningioma (30.9%) ranked first among the preoperative misdiagnosed diseases, similar to what has been stated in other articles; in terms of radiological presentations, IIMT resembles meningiomas [16], including enhancing features and the typical meningeal tail sign. Among all the cases listed in Table 2, there was no instance in which IIMT could be conclusively diagnosed solely based on the radiological features. This observation further underscores the intricate nature of diagnosing IIMT.

## 4. Discussion

IMT represents a rare neoplasm with uncertain biological behavior and underlying pathogenesis. Despite the formal definition of IMT by the WHO in 2002, certain articles published after this date still lack a clear differentiation between IMT, inflammatory pseudotumor, and other histologically akin lesions, a point also noted by Ishihara, Denis, and Vidrine [11,17,18] et al. The majority of the IIMT showed unique MRI performance: iso or low signal on TIWI and T2WI, which may be attributed to the absence of free water and mobile protons [13]. Additionally, IIMT exhibited prominent enhancement and low signal intensity on DWI. These MRI findings were different from other common intracranial parenchymal tumors, which typically present as low-intensity lesions on T1WI and high-intensity lesions on T2WI. This distinctive feature of IIMT is noteworthy in the context of radiological differentiation. In our patient diagnosed with IIMT, the T1WI and T2WI signals predominantly appeared low; however, a small area of high signal intensity was observed within the lesion. The postoperative pathology confirmed the presence of minor hemorrhage, marking the first reported case of IIMT with hemorrhage. The lesion was situated at the right cerebellopontine angle and demonstrated relatively well-defined characteristics. Consequently, the preoperative assessment by the neuroradiologist considered a right cerebellopontine angle acoustic neuroma with hemorrhage. It is noteworthy that the tumor’s atypical T1WI and T2WI signals are distinct from the usual acoustic neuroma. Considering this unique imaging characteristic, the possibility of IIMT could be contemplated, especially if there had been previous IIMT-related studies. In addition to the radiological features, the two main clinical manifestations of headache and seizures might be informative for the diagnosis of IIMT, which has been mentioned in previous reports [13,19].

In the context of this study, the panel of immune markers, including CD34, CD31, ERG, Ki-67, SMA, Vimentin, and others, assumes a pivotal role in the elucidation of both the histological attributes and the molecular profile of IIMT. CD34 and CD31 recognized endothelial markers play a crucial role in the identification of vascular constituents within the lesion [20]. The presence of positive staining for these markers indicates the presence of blood vessels and provides insight into the vascular architecture within the tumor mass. ERG, functioning as an endothelial transcription factor, further contributes to a refined comprehension of the vascular structures and the evaluation of tumor vascularity [20]. Affirmative ERG staining provides insights into the differentiation and arrangement of endothelial cells within the tumor tissue. Ki-67, a representative marker of proliferation, quantifies the proliferative activity of tumor cells. Its expression level, especially within focal hotspot regions, furnishes valuable data regarding the tumor’s growth rate and potential aggressive behavior [21]. SMA, or smooth muscle actin, denotes the presence of smooth muscle cells or myofibroblasts [22]. The scattered yet discernible weak positivity observed in the vascular regions imparts insights into the cellular composition and accentuates the presence of the smooth muscle component within the tumor. Vimentin, serving as a notable mesenchymal marker, signifies the mesenchymal origin of the tumor cells, facilitating their differentiation from other cell types [23]. The collective contribution of these immune markers contributes substantively to the comprehensive characterization of IMT, facilitating the precise diagnosis and comprehension of their biological behavior, and furnishing potential insights relevant to their management and treatment strategies.

*ALK* gene expression is suggestive of the development and recurrence of IIMT [17]; chemotherapeutic agents targeting *ALK* gene inhibition have also been used for the treatment of IIMT with good results [11]. Although some clinicians suggest that surgery is the preferred treatment option for IIMT [24], glucocorticoids have been successful in the treatment of IIMT in some cases [25,26]. Hence, a clear preoperative diagnosis of IIMT would allow for the possibility of an initial steroid-based therapeutic approach, potentially alleviating patient discomfort and reducing postoperative complications, should it prove effective.

**Table 2 diagnostics-13-02725-t002:** Information about inflammatory myofibroblastic tumor.

Patient [Ref]	Location	Complaints	Meningeal Tail Sign	Metastasis	Treatment	Radiological Manifestations	Misdiagnosis	Recurrence
17/M [12]	left posterior fossa	headache	NA	➖	S	obviously enhancing mass	meningioma	➖
6/M [27]	bitemporal lobe	deafness, right hemiparesis, and bilateral cerebellar signs	➖	➖	S	CT: obviously enhancing mass	NA	no recurrence at 6 years
14/F [28]	right posteromedial frontal lobe	headache	➖	➖	S+H+R	obviously enhancing mass	meningioma	recurrence at 6 months
48/F [29]	hypothalamus	drowsiness, hyperthermia, vomiting, and headache	➖	➖	NA	NA	NA	➖
19/F [29]	hypothalamus	visual loss, headaches, and drowsiness	➖	➖	S+R	NA	NA	NA
16/M [30]	right frontoparietal convexity	progressive weakness in the left leg	NA	➖	S	CT: iso-density mass	meningioma	➖
77/F [31]	left frontal region and clival	dementia, urinary incontinence, and anorexia	➖	NA	H	low SI on both T2WI and T1WI, homogeneous enhancement	chronic subdural hematoma	➖
40/M [32]	left trigeminal nerve and left cavernous sinus	decrease in visual acuity or loss of visual field	➖	➖	S+H	T1WI: iso-SI, obviously enhancing	NA	➖
57/M [33]	on the falx cerebri in the frontal area	headache	➖	➖	S	NA	NA	NA
56/M [34]	pituitary stalk	frontal cephalalgias	NA	➖	S	T1WI: iso-SI, obviously enhancing	NA	➖
30/M [34]	left cavernous sinus and tentorium cerebelli	parieto-occipital cephalalgias	NA	➖	S	T1WI: low-SI	NA	➖
11/M [34]	vermis cerebelli	occipital cephalalgias	NA	➖	S	T1WI: low-SI; T2WI: low-SI at the edge of the lesion and iso-SI in the center	NA	➖
40/M [34]	right cavernous sinus	right fronto-orbicular cephalalgias	NA	➖	S	CT: obviously enhancing; TIWI, T2WI: high-SI	meningioma	recurrence at 2 years
11/M [35]	left frontal lobe	mild headache and nausea	➖	➖	S	T1WI: slightly high-SI T2WI: low-SI, heterogeneous enhancement	NA	➖
60/F [36]	near the anterior tip of the temporal lobe	headache, grand mal seizure, and postictal confusion	➕	➖	S+R	T1WI, T2WI: low-SI, obviously enhancing	NA	➖
57/F [37]	right cerebellopontine angle	right-sided ptosis and diplopia	NA	➖	S+R	NA	NA	NA
17/F [1]	left frontal	left frontal headache	NA	➖	S	NA	NA	recurrence at 2 years
8/M [1]	left temporal	seizure	NA	➖	S	NA	NA	no recurrence at 5 years
15/M [1]	left occipital	right-sided epileptic seizure	NA	intracranial MT	S+R	obviously enhancing mass	NA	recurrence at 6 months
18/F [38]	right temporal region	headache with occasional vomiting and blurring of vision	➖	➖	S	CT: obviously enhancing mass	meningioma	➖
23/M [39]	right parieto-occipital	seizures	➕	➖	S+H	T1WI: low-SI T2WI: high-SI slightly enhancing	meningioma	no recurrence at 3 years
13/M [19]	right frontal lobe	seizures	➖	➖	S	T2WI: hypo-intense Obviously enhancing	NA	no recurrence at 6 months
62/M [40]	right fronto-parietooccipital and falx	focal motor seizures and right-sided tinnitus	➕	➖	S+R	T1WI: iso-SI T2WI: low-SI obviously enhancing	meningioma	➖
6/M [41]	right parietal region	seizure	➖	➖	S	obviously enhancing	NA	➖
41/M [42]	right occipital lobe	epileptic seizure	➖	intracranial MT	S	obviously enhancing	High grade glioma, brain metastasis tumor	recurrence at 11 years
22/M [43]	under surface of left tentorium	headache	➕	➖	S	obviously enhancing	meningioma	no recurrence at 19 months
64/F [44]	beside the anterior parasagittal region	headache	NA	➖	S	obviously enhancing	meningioma	no recurrence at 3 years
70/M [45]	cranial base, frontal region, floor of the third ventricle	progressive visual disturbance	➖	NA	H+R	obviously enhancing	pituitary tumor, chordoma, plasmacytoma	NA
36/F [42]	left cerebellopontine angle	left-sided headache, tinnitus, and hearing loss	NA	MT to cervical	S+R	obviously enhancing	meningioma	recurrence at 1.5 years
52/M [46]	left cerebellopontine angle	decreased sensation on the left side of the face, hearing loss, headache, and vomiting	➖	➖	S	T1WI: iso-SI, T2WI: low-SI heterogeneous enhancement	trigeminal neurinoma, meningioma	no recurrence at 6 months
30/F [47]	left temporo-parietal extra parenchymal	worsening headache and memory disturbance	NA	➖	S	CT: iso-density mass obviously enhancing	subacute subdural hematoma	➖
48/F [47]	left temporal lobe	headache	NA	➖	S	T1: low-SI, T2: high-SI obviously enhancing	NA	➖
14/F [48]	cavernous sinus, right middle cranial fossa, pterygopalatine, and infratemporal	headache	➖	➖	H+methotrexate +6-mercaptopurine	NA	NA	recurrence at 18 months
63/M [49]	right frontal lobe	progressive left hemiparesisi	➖	➖	S	MRI: the solid part of the mass was significantly enhanced	NA	➖
18/F [50]	left frontoparietal	generalized seizure	➖	➖	S	CT: calcifications and extensive ossification T1: low-SI	NA	➖
58/F [51]	left fronto-temporal	headache	➖	➖	S+H	obviously enhancing mass	NA	no recurrence at 18 months
47/M [52]	right cerebellum-pontine angle	reduced visual acuity, hearing loss, difficulty in walking, and urinary retention	NA	NA	H+R	T1WI: iso-SI, T2WI: low-SI	NA	NA
26/F [53]	left frontotemporal region	severe headache, left eye discomfort with diplopia, left ear pain	NA	NA	high-dose dexamethasone and thalidomide	MRI: extensive enhancing, dural thickening over left frontotemporal lobes	NA	NA
52/F [53]	right lateral ventricle	slipped and fell with head injury	NA	NA	S	NA	NA	NA
45/M [53]	right frontal region	progressive left-sided weakness	NA	NA	S	CT: right frontal dural based tumor with peritumoral edema	NA	NA
51/M [26]	right cerebellopontine angle	vertigo, diplopia, headache and fibrillationa of the tongue	NA	➖	H	obviously enhancing mass	NA	recurrence at 7 years
56/M [54]	left basal ganglia	headaches and right-sided weakness	➖	➖	H+biopsy	TIWI: low-signal, heterogeneous enhancement	NA	➖
60/M [55]	right cerebral hemisphere	gait disturbance and ataxia	➖	➖	S	T1WI, T2WI: iso-SI, obviously enhancing	NA	➖
82/F [56]	right temporal region	headache and memory decrease	➕	➖	S	T1WI: slightly high-SI homogeneous enhancement	meningioma	no recurrence at 13 months
38/M [57]	right mastoid	blurred vision and headache	➖	➖	S+H	MRI: obviously enhancing CT: bone erosion around the tumor MRV: right sigmoid sinus occlusion	NA	➖
48/F [15]	right temporal region	depression, paranoid personality, and memory impairment	➖	➖	S	T2WI: iso-SI, DWI: not limited, obviously enhancing	pleomorphic xanthoastrocytoma, cystic meningioma, cystic glioma	➖
20/F [58]	bilateral temporal regions	headache	NA	➖	S	T1WI: low-SI T2WI: mixed-SI CT: iso-density mass heterogeneous enhancement	NA	recurrence at 7 months
10/M [2]	left transverse-sigmoid junction	mastoid tenderness and headache	➕	➖	S	CT: obviously enhancing	meningioma	➖
15/M [59]	left parietal region	headache, vomiting, and lethargy	➖	➖	S	T1WI: iso-SI T2WI: low-SI obviously enhancing	NA	NA
46/F [14]	right temporal lobe and right cerebellar hemisphere	headache, unstable walking	NA	NA	S+H	T1WI: iso-SI T2WI: low-SI MRV: right transverse sinus and sigmoid sinus occlusion	malignant meningioma	recurrence at 2.5 years
21/F [5]	right frontal lobe	bilateral blurred vision	➖	➖	S	T1WI, T2WI: low-SI obviously enhancing	meningioma	➖
54/M [14]	right frontal lobe	headache	➕	NA	S	T1WI: iso-SI T2WI: low-SI obviously enhancing	NA	➖
27/M [16]	right frontal parietal region	seizures	➕	NA	S	obviously enhancing	meningioma	NA
80/F [60]	left choroidal fissure between the amygdala and cerebral peduncle	headache, dizziness, confusion, and gait instability	➖	➖	S	obviously enhancing	aneurysm	➖

NA = non-available; S = surgery; H = hormones; R = radiotherapy; NSAIDs = non-steroidal anti-inflammatory drugs; CT = computed tomography; MRI = magnetic resonance imaging; T1WI = T1-weighted imaging; T2 = T2-weighted imaging; SI = signal intensity; MRV = magnetic resonance venography; DWI = diffusion-weighted imaging. ➕ = positive; ➖ = negative.

## 5. Conclusions

This study presents the inaugural case of IIMT with internal hemorrhage and undertakes an all-encompassing review of pertinent literature accessible in the PubMed database. While the direct diagnosis of IIMT presents challenges, considering IIMT as a plausible differential diagnosis emerges as a feasible avenue. This approach assumes substantial significance in steering treatment strategies effectively.

## Figures and Tables

**Figure 1 diagnostics-13-02725-f001:**
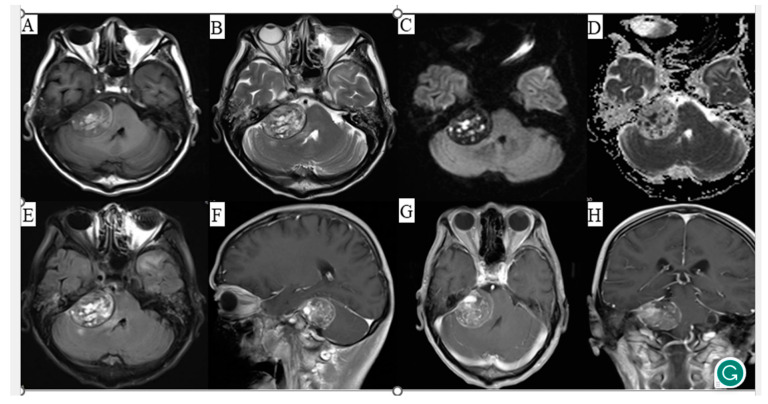
Magnetic resonance imaging (MRI) of the patient’s head with corresponding image sequences. (**A**) T1-weighted imaging, (**B**) T2-weighted imaging, (**C**) Diffusion-weighted imaging, (**D**) Apparent diffusion coefficient, (**E**) Fluid attenuated inversion recovery, (**F**) Sagittal contrast-enhanced scan, (**G**) Axial contrast-enhanced scan, (**H**) Coronal contrast-enhanced scan.

**Figure 2 diagnostics-13-02725-f002:**
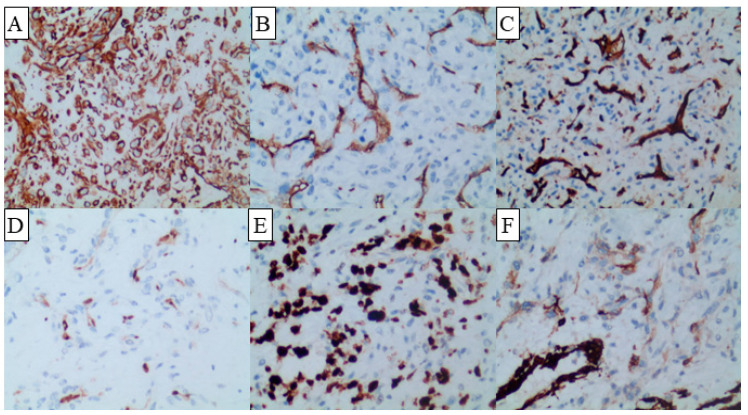
Immunohistochemical staining of the lesion demonstrated positive reactivity for vimentin (**A**) throughout and revealed scattered weak positivity for SMA (**B**) in the vascular regions. Additionally, the vascular regions exhibited positive staining for CD31 (**C**), ERG (**D**), Ki-67 (**E**), and CD34 (**F**). The original magnification is 20X.

**Table 1 diagnostics-13-02725-t001:** WHO classification of fibroblastic/myofibroblastic and some representative immunomarkers.

Type of Tumor	Representative Immunomarkers
Nodular fasciitis	CD34 (+), SMA (+), Desmin (−), S100 (−), ALK-1 (−), Ki-67(+)
Proliferative fasciitis	SMA (+), Desmin (−), CD34 (+), S100 (−), ALK-1 (−), Ki-67 (+)
Proliferative myositis	SMA (+), CD34 (+), Desmin (−), S100 (−), ALK-1 (−), Ki-67 (+)
Myositis ossificans	SMA (+), Desmin (−), CD34 (−), S100(−), ALK-1 (−), Ki-67 (−)
Ischaemic fasciitis	SMA (+), Desmin (−), CD34 (+), S100(−), ALK-1 (−), Ki-67 (+)
Elastofibroma	SMA (+), Desmin (−), CD34 (−), S100 (−), ALK-1 (−), Ki-67 (−)
Fibrous hamartoma of infancy	SMA (+), Desmin (−), CD34 (+), S100(−), ALK-1 (−), Ki-67 (+)
Myofibroma/Myofibromatosis	SMA (+), Desmin (−), CD34 (−), S100 (−), ALK-1 (−), Ki-67 (+)
Fibromatosis colli	SMA (+), Desmin (−), CD34 (−), S100 (−), ALK-1 (−), Ki-67 (−)
Juvenile hyaline fibromatosis	Vimentin (+), CD34 (−), CD68 (−), SMA (−),S 100(−), Factor XIIIa (−)
Inclusion body fibromatosis	Vimentin (+), CD34 (−), CD68 (−), SMA (−), S100 (−), Desmin (−)
Fibroma of tendon sheath	Vimentin (+), CD34 (−), Desmin (−), SMA (−), S100 (−), CD68 (−)
Desmoplastic fibroblastoma	Vimentin (+), CD34 (−), Desmin (−), SMA (−), S100(−), CD68 (−)
Mammary-type myofibroblastoma	Vimentin (+), CD34 (−), Desmin (−), SMA (+), S100 (−), CD68 (−)
Calcifying aponeurotic fibroma	Vimentin (+), CD34 (−), Desmin (−), SMA (−), S100 (−), CD68 (−)
Angiomyofibroblastoma	Vimentin (+), CD34 (−), Desmin (−), SMA (+), S100 (−), CD68 (−)
Cellular angiofibroma	Vimentin (+), CD34 (+), Desmin (−), SMA (−), S100 (−), CD68 (−)
Nuchal-type fibroma	Vimentin (+), CD34 (−), Desmin (−), SMA (−), S100 (−), CD68 (−)
Gardner fibroma	Vimentin (+), CD34 (−), Desmin (−), SMA (−), S100 (−), CD68 (−)
Calcifying fibrous tumor	Vimentin (+), CD34 (−), Desmin (−), SMA (−), S100 (−), CD68 (−)
Giant cell angiofibroma	Vimentin (+), CD34 (−), Desmin (−), SMA (+), S100 (−), CD68 (−)
Superficial fibromatoses (palmar/plantar)	SMA (+), CD34 (−), Desmin (−), S100 (−), Beta-catenin(CTNNB1) (+)
Desmoid-type fibromatoses	SMA (+), CD34 (−), Desmin (−), S100 (−), Beta-catenin(CTNNB1) (+)
Lipofibromatosis	CD34 (+), SMA (+), Desmin (−), S100 (−), Beta-catenin(CTNNB1) (−)
Solitary fibrous tumor	CD34 (+), CD99 (+), Bcl-2 (+), CD31 (−), S100 (−), Desmin (−)
Inflammatory myofibroblastic tumor	ALK [partly(+)], SMA (+), CD34 (+), Desmin (−), CD117 (−), S100 (−)
Low grade myofibroblastic sarcoma	SMA (+), Desmin (−), CD34 (−), CD117 (−), ALK (−), S100 (−)
Myxoinflammatory	CD68 (+), CD163 (+), SMA (+), Desmin (−), CD34 (−), S100 (−)
Infantile fibrosarcoma	SMA (+), Desmin (−), CD34 (−), CD99 (−), ALK (−), S100 (−)
Adult fibrosarcoma	SMA (+), CD34 (−), Desmin (−), S100 (−), EMA (−), CD99 (−)
Myxofibrosarcoma	CD34 (−), SMA (−), Desmin (−), S100 (−), EMA (−), CD99 (−)
Low grade fibromyxoid sarcoma	CD34 (−), SMA (−), Desmin (−), S100 (−), EMA (−), MUC4 (+)
Sclerosing epithelioid fibrasarcoma	CD34 (−), SMA (−), Desmin (−), S100 (−), EMA (−), INI1(SMARCB1):(+)

Ki-67: Ki-67 Antigen; Factor XIIIa: Factor XIIIa (F13A1); Beta-catenin (CTNNB1): Beta-Catenin; INI1 (SMARCB1): SWI/SNF-related matrix-associated actin dependent regulator of chromatin subfamily B member 1; Bcl-2: B-Cell Lymphoma 2.

## Data Availability

All of the data in this study can be found in the PubMed database.

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
