# Peer review of "Intracranial Inflammatory Myofibroblastic Tumor: A Literature Review and a Rare Case Misdiagnosed as Acoustic Neuroma"

_diagnostics, 2023, doi:10.3390/diagnostics13172725_

Round 1

Reviewer 1 Report

This is a nicely written report with a great case presentation and literature review.

Comment:

Please explain to the general medical audience the purpose of the immune markers used. What CD34, CD31, ERG, Ki-67, SMA, Vim, etc. are for?  Materials and methods would be a good place for it. The discussion may be a good place for positive markers to incorporate and overview. 

Reviewer 2 Report

1-      The references 3-7 mentioned in the sentence on line 46 need to be updated to their latest versions.

2-      Similarly, references 11-17 on line 54 should also be updated to their most current editions.

3-      A reference should be added to support the statement on lines 57-58 regarding the histological aspects of IMT. Please include a source that elaborates on this topic.

4-      Correct the typo in line 93: Change "T-weighted imaging" to "T2-weighted imaging."

5-      Clarify lines 114 and 115 by referring to a specific classification guideline, such as WHO,and present the information through a table or flowchart format to enhance understanding.

6-      Please rephrase the introductory paragraph on page 7 (lines 181-187) for greater clarity and coherence.

Minor editing of English language required.
